# Rheological, Thermal, Superficial, and Morphological Properties of Thermoplastic Achira Starch Modified with Lactic Acid and Oleic Acid

**DOI:** 10.3390/molecules24244433

**Published:** 2019-12-04

**Authors:** Carolina Caicedo, Rocío Yaneli Aguirre Loredo, Abril Fonseca García, Omar Hernán Ossa, Aldo Vázquez Arce, Heidy Lorena Calambás Pulgarin, Yenny Ávila Torres

**Affiliations:** 1Grupo de Investigación en Desarrollo de Materiales y Productos, Centro Nacional de Asistencia Técnica a la Industria (ASTIN), SENA, Cali 760003, Colombia; omar.ossa@sena.edu.co (O.H.O.); avazqueza@sena.edu.co (A.V.A.); hcalambas@sena.edu.co (H.L.C.P.); 2Consejo Nacional de Ciencia y Tecnología (CONACYT)-Centro de Investigación en Química Aplicada (CIQA), Blvd. Enrique Reyna Hermosillo 140, Saltillo, Coahuila 25294, Mexico; yaneli.aguirre@ciqa.edu.mx (R.Y.A.L.); abril.fonseca@ciqa.edu.mx (A.F.G.); 3QUIBIO, Facultad de Ciencias Básicas, Universidad Santiago de Cali, Pampalinda, Santiago de Cali 760035, Colombia; yennytorres@usc.edu.co

**Keywords:** thermoplastic starch, oleic acid, lactic acid, thermal properties, contact angle

## Abstract

The modification of achira starch a thermoplastic biopolymer is shown. Glycerol and sorbitol, common plasticizers, were used in the molten state with organic acids such as oleic acid and lactic acid obtaining thermodynamically more stable products. The proportion of starch:plasticizer was 70:30, and the acid agent was added in portions from 3%, 6%, and 9% by weight. These mixtures were obtained in a torque rheometer for 10 min at 130 °C. The lactic acid managed to efficiently promote the gelatinization process by increasing the available polar sites towards the surface of the material; as a result, there were lower values in the contact angle, these results were corroborated with the analysis performed by differential scanning calorimetry and X-ray diffraction. The results derived from oscillatory rheological analysis had a viscous behavior in the thermoplastic starch samples and with the presence of acids; this behavior favors the transitions from viscous to elastic. The mixture of sorbitol or glycerol with lactic acid promoted lower values of the loss module, the storage module, and the complex viscosity, which means lower residual energy in the transition of the viscous state to the elastic state; this allows the compounds to be scaled to conventional polymer transformation processes.

## 1. Introduction

Nowadays, the use of plastic materials derived from petroleum has generated a negative environmental impact [1,2]; from this reason was born the necessity to use polymers obtained from renewable sources as a great alternative as packaging materials in single-use products, due to their biodegradation or compostability [3,4]. According to the latest market data collected by European Bioplastics and Nova-Institute [5], the global production capacity of bio-based plastic packaging materials will increase by 2022 in 20% (2.45 million tons) [6,7]. Some biopolymers such as cellulose, poly (lactic acid), starch, chitosan, and poly (hydroxyalkanoates) have been modified aiming to develop barriers to various gases, moisture, and even microorganisms [8]. Granular, gelatinized, and plasticized starch has been used for the generation of packaging films, biodegradable containers, stabilizers, industrial and engineering adhesives, etc. [9,10].

The manufacture of materials based in starch represents an important potential in the market due to the wide availability in its raw material and the versatility to make chemical modifications [11]. Among the sources of starch, of which studies related to thermoplastification have been carried out, are: Corn [12], cassava [13], potato [14], rye [15], banana [16], wheat [1,2], rice [17], barley [18], amaranth [19], chia [20], achira [21], among others [22]. The achira is a plant widely cultivated in different countries of Latin America and Asia [23,24], and it has become a viable resource, mainly in Colombia where there are industrial of starch extraction plants, which has interesting physicochemical properties to develop plastic films [25].

This has been demonstrated by Andrade-Mahecha et al. [21] who evaluated achira-based thermoplastic starch (TPS) with different proportions of glycerol (from 15% to 35% *w*/*w*) and they concluded that high glycerol content contributes to greater flexibility and reduces the opacity of films. Furthermore, the conditions of the casting method used, such as process temperature (90 °C) and drying temperature (50 °C), favor the preparation of films with greater resistance and stiffness, while a low relative humidity (30%) significantly increases stiffness and reduces solubility.

Other studies have focused on understanding the influence of plasticizers, such as sorbitol and glycerol, which have been widely used in these systems. Jafarzadeh et al. [26] demonstrated a relationship between plasticizer contents in semolina films and the barrier properties to UV light, permeability to water vapor, and oxygen. However, the increase in plasticizer concentration (up to 50%) improved flexibility, decreased resistance, and caused the film to yellow. On the other hand, studies have been carried out on the incorporation of chemical agents such as lubricants, fiber nanoparticles, and others, in order to improve processability conditions and thermo-mechanical properties in the final material [27].

The starch can be modified by oxidation [28], esterification [29], and etherification [30], which favors adhesion, film formation with greater resistance and decreased viscosity, retrodegradation, and hydrophilic character. Several studies add to this disturbing problem, such as Saliu et al. [31] (2018), who proposed a new method in situ, which is carried out in two steps: The first is the intramolecular union of starch citrate, and the second consists of obtaining a complex of ammonium acetate and thiourea, which significantly improves the resistance to water vapor and oxygen of thermoplastic starch. López et al. [32], in 2013, obtained biodegradable films with potential application as a container for the control of plant respiration and senescence from native and acetylated corn starches using the blown extrusion technique. Similarly, studies of acetylation and stertification for starch with acetic and maleic anhydride are reported to obtain foams that replace expanded polystyrene [33]. Murillo et al. [34] developed a hyperbranched polyester polyol that acts as a plasticizer, having been evaluated by means of functionalization in cassava starch; these results demonstrated its potential as a plasticizing agent [35]. On the other hand, the effect of tartaric acid on TPS and compounds of PBTA/TPS has been reported, which in a concentration of ≤1%, contributes as a catalyst, coupling agent, and significantly improves the mechanical, hydrophobic, as well as velocity properties of degradation given these properties. However, it is surprising that, to date, no studies have been conducted on functionalization with acidic agents that promote a change in hydrophilic properties using achira starch [36]. 

The main objective of this work was to obtain mixtures of functionalized starch through to reach better effectiveness in the gelatinization process to processability conditions by the molten state for possible applications as raw material in the packaging sector. Accordingly, in this purpose, the structural, morphological, rheological, and thermal properties were evaluated and the hydrophilic properties of achira starch were determined using common plasticizers (glycerol and sorbitol) with different proportions of oleic acid and lactic acid. 

## 2. Results

The concentration of plasticizer was defined in a preliminary test, as the minimum amount necessary to obtain manageable samples; our research group developed starch-based materials such as potato and corn, where the amount of glycerol needed for the formulation of the materials has been found to be 25% to 30% [37,38], even though these organic acids could function as plasticizers. In the literature, various starch:plasticizer ratios are reported that vary between 90:10 and 50:50 [38,39]; however, the ratio of these mixtures when applied in conventional processes, such as extrusion, is limited to 80:20 and 70:30 [40,41,42]. Taking into account the above, it was set at a ratio of 70:30 and only organic acids were varied in order to know the effect on the rheological, surface, and thermal properties that organic acids can cause.

### 2.1. Rheometric Analysis

The behavior of the mixtures during processing is shown in Table 1. The maximum torque ratio for TPS samples with sorbitol as a plasticizer in the presence of oleic acid showed a significant decrease in stabilization with respect to the blank. However, the TPSs O-9 mixture resulted in higher values, probably due to an insufficient interaction of oleic acid with the other components, which was visualized by migration of the acid in the final material. Likewise, the TPS samples with glycerol as a plasticizer had, in all cases, lower values with respect to the target, and the best behavior was achieved with minimal additions of the acidic agents (3% and 6%). In general, the minimum temperatures were maintained in a range between 85 and 114 °C, and the process energy was proportional to the torque, resulting in more favorable energy consumption for samples with lactic acid that showed a decrease of up to 84% in the maximum torque and up to 92% in the stabilization torque. In all cases, as the concentration of organic acids increased to a concentration of 6%, the process softened; however, it should be noted that the changes in the response variables (torque, temperature, and energy) with the 9% incorporation were not significant, so it was concluded that for this starch mixing process, the organic acid conditions should not be. With the above, it could be concluded that mixtures for process effects do not require additions greater than 6%.

### 2.2. Rheological Analysis

The curves corresponding to the storage and loss modules were analyzed according to the angular frequency for the target samples in this study, which are presented in Figure 1 and Figure 2. In general, these exhibit an increase in the value of the modules as the angular frequency increases. TPS samples were viscous; this behavior was observed in the studied frequency range of the material between 0.1 and 630 rad/s. However, TPS samples with sorbitol and lactic acid showed a mainly elastic behavior (without transitions), while the incorporation of oleic acid generated viscous to elastic transitions with low modules. TPS samples with glycerol as a plasticizer in the presence of oleic acid, unlike lactic acid, showed viscous to elastic transitions at lower frequencies; however, in these samples, viscous behavior predominates. The complex viscosity values obtained (Figure 1 and Figure 2, Appendix A) for the mixtures showed dependence on the type of acid. For lactic acid, they decreased with the increase thereof, while fatty acid increased the viscosity values twice as much. It is interesting to observe the influence of the type of plasticizer, because sorbitol showed higher values than glycerol as expected. In general, these polymers show a strong effect of shear thinning; this dependence on the frequency range studied was very representative. The reduction in the viscosity of the TPS corresponds to the dissociation of the interactions and disentanglement of the double helices of amylose and amylopectin. On the other hand, at angular frequency of ~100 rad/s, a considerable increase in the η* value was observed for all TPS with lactic acid (shear thickening). This behavior has been evidenced in tapioca starch samples possibly given by the formation of a structure of the elastic network, which occurs when the material is exposed to external stresses due to a large increase in the dispersion’s storage modulus [43]. The formation of this structure can be produced by chemical or physical crosslinking, the complexation reaction between amylose and lipids, or the physical entanglement of the high molecular weight polysaccharides [44]. The angular frequency value, in which the steep increase appears in η*, did not show a trend with the proportions of starch and acid. Starch granules were more dispersed in the mixture of lactic acid and plasticizer. As a consequence of the cut, these separated, but their interaction was lower, suggesting low viscosity. In fact, the interactions between lactic acid/plasticizer and starch were lower for these samples, because a large percentage of acid does not interact with starch.

### 2.3. IR Analysis

The infrared spectroscopy analysis of the mixtures allowed us to identify the interaction of the hydrogen bond of the TPS with lactic acid and oleic acid (see Appendix A), and the characteristic absorption bands of the CH groups of -CH_2_- in C_6_ from the glucose unit at 2925 cm^−1^ and 1450 cm^−1^. Meanwhile, the absorption band around 1650 cm^−1^ belongs to the flexural vibration of H_2_O attributed to the absorption of water by hydroxyl groups in the starch. A characteristic absorption band at 1740 cm^−1^ is assigned to C=O for acid groups, which is most notable with high content (9%) of oleic and lactic acid. Other bands located at 1200–1300 cm^−1^ and 1000–1200 cm^−1^ were assigned to C-O-C stretching and C-O-H flexion, respectively [16,43]. Other characteristic bands at 1080 cm^−1^ and 1157 cm^−1^ indicate the stretching of the C-O bond of the C-O-H group in mixtures of TPS with oleic and lactic acid. The band of 1080 cm^-1^ shifted to 1078 cm^−1^ and 1150 cm^−1^ to 1148 cm^−1^, indicating that the starch OH group was involved in the formation of hydrogen bonds. The vibrations at 999 cm^−1^ and 1020 cm^−1^ are attributed to the stretching of the C-O bond of the C-O-C group. The bands moved from 998 cm^−1^ to 995 cm^−1^ and from 1018 cm^−1^ to 1016 cm^−1^, respectively; showing that the incorporation of organic acids helps the formation of more stable hydrogen bonds than those obtained with pure plasticizers. The band observed at 925 cm^−1^ is associated with the vibration of the glycosidic bonds of starch, which was observed to be higher in the films plasticized with glycerol compared to the other plasticizer, while the use of the hydrophobic component seems not had an effect on the size of the observed band.

### 2.4. SEM Analysis

Figure 3 shows the images captured with an SEM of the TPS modified with the two acids studied at a concentration of 6% and with the two plasticizers, and these images provide information on the general characteristics of the fracture surface of the materials. Micrographs for TPS samples in the absence of organic acids reveal a low destructuring of the starch granules. The incorporation of lactic acid generates a material with a compact appearance, and a reduction in the size and content of the non-gelatinized starch granules. On the other hand, with the presence of oleic acid, more compact surface materials are obtained that allow the interconnection between domains with the presence of less porous interfaces. In general, the results obtained with the SEM confirmed a better interaction of the plasticizer with the starch in the samples with the acid agents, and this result was corroborated with the analysis performed by DSC and XRD. The increase in the concentration of the acid agent allows a high dispersion of the starch; likewise, it is possible that the effect of the acids into the materials is mainly as a dispersing agent, which generates a high level of compactness and a regular and homogeneous surface morphology.

### 2.5. Thermal Analysis

The results of the thermogravimetric analysis are presented in Figure 4, and the thermograms showed that the plasticized samples with sorbitol had higher thermal stability compared with plasticized samples with glycerol. However, both plasticizers, sorbitol and glycerol, afford higher thermal stability with respect to pristine achira starch, which lost the weight for T_10_ at 96 °C (Appendix A). The sample with the highest thermal stability corresponded to the sample with sorbitol and oleic acid at 9%, which showed the T_10_ at 255.6 °C. Adversely, the formulation of TPSs with 9% of lactic acid generated the lowest T_10_ at 101.4 °C below of the blend of achira starch with the glycerol, which showed the T_10_ at 129.9 °C.

In general, the maximum degradation for the TPSs and TPSg samples was ~304 °C; this result was obtained with the values of the first derivative of the thermogravimetric analysis (Appendix A). For TPSg the degradation temperatures were lower and with a continuous loss of the most volatile constituents (acids and plasticizers) about 30%, which reflects less interaction with starch. The behavior of the mixtures is largely linked to the type of plasticizer used. Samples of TPSs are better favored with oleic acid, while in TPSg, this behavior was presented with lactic acid, with which it could be deduced that an organic acid has better interaction with a specific plasticizer. This knowledge could be of help for the elaboration of future materials in which these fatty acids are to be incorporated, knowing with which plasticizer it is more convenient to mix it in this type of material.

Regarding melting temperatures and enthalpies (Figure 5), it was observed that TPSg samples favored the transformation process by presenting a lower heat requirement compared to TPSs. The melting temperatures and enthalpies will be taken from the first melting cycle. The incorporation of lactic acid is an agent that achieves a more radical change in the heat absorption of the solid–liquid transition than oleic acid. This could be explained by the size of the structure and steric impediment that the oleic acid generates in its defect, such is the case of the TPSg sample with 6% oleic acid; the presence of two widened transitions that could be interpreted as weak interaction is observed with the starch. An indication of the interaction between the material and the plasticizer is the decrease in the crystallinity of the sample, which is observed as low enthalpy values (≤ 10 J/g). The glass transition temperature (T_g_) was not calculated in this research; however, it is a parameter of great importance that will be determined in future research, since the T_g_ is modified by the presence of plasticizing agents, as well as by the modifications that are made to the starches. Mali and collaborators [44] determined the change in the T_g_ values of films made from different starches, such as corn, cassava, and yam, where they found that the increase in glycerol concentration significantly modified the T_g_ of the materials, in which it was reduced from 54 to 21 °C, 56 to 27 °C, and 53 to 19 °C, respectively, when the percentage of plasticizer increased from 0% to 40%. It is important to mention that in case of a correct reaction between starch and plasticizer, the peak indicating the melting point of the pure plasticizer (17 °C and 96 °C for glycerol and sorbitol, respectively) should not appear in the DSC analysis since this constituent would become part of the structure of the TPS. As we can see, in some cases, the TPSg samples record a large endothermic peak between 40 °C and 100 °C representing the evaporation of glycerol that was not incorporated. Thus, in the joint analysis of TGA and DSC, the loss of mass is evidenced by partial volatilization of the plasticizer that fails to interact. In the TPSs, fusion transitions are observed between 120 °C and 190 °C, which varied depending on the proportions of lactic acid but independent for oleic acid, probably due to lower efficiency in incompatibility, and the nature of fatty acids versus polar molecules.

### 2.6. X-Ray Diffraction (XRD) Analysis

The XRD patterns of the materials studied are shown in Figure 6; both TPSs and TPSg samples and those derived with acids show peaks around 13.1°, 17°, 18.6°, 20°, and 22°. These peaks are characteristic of a crystalline structure of type “B” starches, which is generally obtained from tuber and roots, such as potato starch and achira starch. The strongest diffraction peak in the XRD pattern appears at 2θ = 20° and there are also some small peaks at 2θ values around 13.1°, 18.6°, 17°, 22°, and 26.7°. Several authors have published similar results for achira starch [45,46,47]. The incorporation of 6% oleic acid in the plasticized materials with glycerol, as with sorbitol, favored the crystallinity of the starch matrix, observing an important increase in the peak at 2θ = 20°. Thitipraphunkul et al. [48] have reported that this type of flour has a large proportion of long branches of amylopectin chains. The peaks at 17.22° and 24° have a decreased intensity. According to Gunaratne and Hoover [49], it has been shown that the type of crystalline polymorph is influenced by the fatty acid content due to the formation of lipid-amylopectin complexes in starch. It is likely that these complexes do not form crystallites of adequate size to provide a test indicating their existence at 5.6° in achira starch. This is because a minimum size of crystalline domains is required for the detection of crystallinity by X-ray diffractometry [50]. It has been reported that the crystallinity of starch films is closely related to the organization of the starch chain [51]. Therefore, our crystallinity results indicated that both sorbitol and glycerol in achira starch films with oleic acid provided a more ordered structural matrix.

### 2.7. Contact Angle Analysis

The contact angle values are presented in Figure 7 and a comparative analysis was performed for each of the TPS with organic acids relative to the target (TPS). The TPS with sorbitol (TPSs) had a value of 34.1°; the maximum reached was achieved with the TPSs O-6 with 38.2° and this increase represented 12.0% over the target. The high wettability (minor contact angle) of achira starch films with lactic acid (L-9) in the presence of both sorbitol and glycerol indicated that the starch matrix in these films had a greater number of sites available for hydrogen bonding with water molecules due to less interaction between hydroxyl groups of starch and plasticizing hydroxyl or amide groups (polar components) during film preparation, behavior similar to that reported for babassu starch films [51]. In general, the contact angle values decreased as the concentration of lactic acid increased, and these results can be associated with the fact that this acid reached a better gelatinization process when the hydrophobic and polar components are oriented to surface of the gel [52]; the steric hindrance is the key to orient the most polar components of the branched-chain of amylopectin. In blends of starch with lactic acid, it is possible that the steric phenome is more abundant than in blends of starch with oleic acid. The good gelatinization in the blend of starch with lactic acid is evidenced in XRD results, where it was shown that the amorphous starches corresponded to samples with lactic acid in sorbitol, and in glycerol. As well as this, the lowest enthalpy energy corresponds to samples with lactic acid, which indicates it is the low atomic order. While the behavior was different with oleic acid, due to greater hydrophobicity, it was both plasticizers in O-6. This result is according to the high crystallinity showed in XRD to Oleic acid.

Samples of TPS with glycerol (TPSg) showed a higher stable trend among the different additions. A contact angle of 43.1° was reached with respect to 31.3° corresponding to the target (TPSg). It was possible to reduce hydrophilicity with the incorporation of the two organic acids used, which can significantly improve the effectiveness of this material as a compatibilizing agent for mixtures between biopolymers, as in the case of TPS with PLA. These results are comparable with TPS films plasticized with glycerol (38.7°) reinforced with oxidized microcrystalline cellulose [53], in which contact angles between 46.1° and 67.9° were measured. Other hydrocolloid-based films such as the casein plasticized with glycerol reported by Chevalier et al. [54] presented similar contact angles (~57°). Generally, carbohydrates that form plastic films have lower contact angles than those obtained in hydrophobic materials such as low-density polyethylene film (93.9°–100.2°) or plexiglass (91.5°) [55]. Hydrophilicity for mixtures with other biopolymers such as PBAT is an interesting factor that represents rapid biodegradation for PBAT/TPS [56]. For interactions of starch-based films plasticized with glycerol and sorbitol in different relative humidity (RH) environments [57], an increase in the values of the water diffusion coefficient and vapor permeability was observed of water and the coefficient of solubility by increasing the concentration of plasticizer and RH.

## 3. Materials and Methods

### 3.1. Materials

Achira starch was supplied by Surtialmidon (Huila, Colombia) with a density of 1.59 g/mL. The plasticizers were obtained from Sigma Aldrich, glycerol with a density of 1.26 g/mL (purity: 99.68%), and sorbitol with a density of 1.28 g/mL (purity: 99.5%). The oleic acid was obtained from Quimifast Ltd.a, and lactic acid was supplied by Mol Labs (purity: 85%), the density of which was 1.21 g/mL.

### 3.2. Preparation of Polymeric Materials

The respective amounts of achira starch; plasticizing agents sorbitol (s) and glycerol (g); and organic acids, oleic (O) and lactic (L), were carried out in the torque rheometer Thermo Scientific equipment HAAKE Rheomix using roller type rotors. The mixer operated at a rotor speed of 50 rpm at a temperature of 130 °C, and the system was kept under mixing for 10 min. The respective amounts of plasticizing’s and organic acids were added according to Table 2. The materials were obtained in duplicate. Fourteen mixtures of thermoplastic starch were obtained, to which each of the different combinations of oleic (O) and lactic acids (L) were added, for which the proportion of the acidic agent was varied (0%, 3%, 6%, and 9% by weight).

The samples were named as follows: [TPSp and A-n], where p is the type of plasticizer (g or s), A is the type of acidic agent, and n is equal to the proportion of organic acid; for example, TPSg L-6 is a TPS obtained from achira starch and glycerol as a plasticizer, with the addition of 6% lactic acid; TPSs O-3 is a TPS obtained with sorbitol as a plasticizer, with the addition of 3% oleic acid; and finally, TPSg is a TPS obtained with glycerol as a plasticizer, without the addition of organic acid.

### 3.3. Characterization

#### 3.3.1. Rheometric Analysis

Rheometric tests were performed on a Thermo Scientific brand HAAKE Rheomix Lab Mixers model on the compounds at different temperatures using Roller type rotors. The calculation of the filling of the chamber was carried out taking into account Equation (1):w_*c*_ = *V*_*n*_ × *ρ_c_* × *f_d_* × *x_m_*(1)
where w_*c*_ is the weight of the compound, *V_n_* is the net volume of the mixing chamber (69 mL), *ρ_c_* is the density of the compound, *f_d_* is the filling factor (70%), and *x_m_* is the mass fraction of the material.

#### 3.3.2. Rheological Analysis

The viscosity was determined by a rotational rheometer (DHR-2, TA Instruments) with controlled stress and parallel plate configuration using the equilibrium flow test. The rheological measurements were performed at 150 °C and the shear rate was in the range of 0.001 to 300 s^−1^. The dynamo-mechanical tests settings were percentage of deformation between 1% and 10%, with a frequency range of 0.1 to 628.10 rad/s, to determine the storage modulus and the loss modulus.

#### 3.3.3. IR Analysis

The spectra of the TPS samples were determined using an Agilent Cary 630 FTIR spectrometer in total attenuated reflectance mode (ATR). The FT-IR spectrum was obtained in a wavelength range of 450 to 4250 cm^−1^ using 16 scans at a resolution of 18 cm^−1^. Three replicates per treatment were collected, and for the analysis, four spectral regions were selected: 800–1150 cm^−1^ in order to determine the presence of groups C−C, C−O, and C−H; 1600–1900 cm^−1^; and the presence of -OH groups at 3000–3600 cm^−1^.

#### 3.3.4. Scanning Electron Microscopy Analysis

The sample was obtained by cryofracture of the TPSs, TPSg mixtures with 6% lactic and oleic acid resulting from the rheometer were examined, and the micrographs were digitally captured using a scanning electron microscope JEOL, model JCM 50000. A voltage of 10 kV was applied. Prior to the tests, both specimens were sputter coated with a layer of gold. Magnifications of 1000× of the fracture surface were taken.

#### 3.3.5. Thermal Analysis

The thermogravimetric (TGA) and differential scanning calorimetry (DSC) analyses were performed on a TGA/DSC 2 STAR System thermogravimetric analyzer, Mettler Toledo. The samples (10 ± 0.5 mg) were placed in alumina crucibles at a heating rate of 20 °C/min using a nitrogen purge at a flow rate of 100 mL/min. In the case of TGA, the analysis was performed from room temperature to 600 °C. The DSC analysis was done at a temperature range between 30 and 300 °C. In this work, the melting enthalpy (Δ*H_m_*) was estimated using the Origin software, the area (A) was determined under the transition curve; these data were entered in Equation (2):(2)$$\Delta H_{m}=\frac{A}{dT/dt}$$

The heating rate is dT/dt and A corresponds to the calculation of the integral of the heat flow (*dt(T)/dt*) along the melting interval (*T_0_*→*T_f_*), as follows in Equation (3):(3)$$A=\int_{T_{0}}^{T_{f}}\frac{dq\left(T\right)}{dt}dT$$

Also, the normalized melt enthalpy of weight is reported.

#### 3.3.6. XRD Analysis

The samples were analyzed in the form of a film in a Rigaku Ultima IV X-ray diffractometer with Cu Kα = 1.54 A, and the voltage and current operation were 40 kV and 44 mA, respectively. The diffractograms were obtained in the interval of Bragg angle (2θ) of 5 and 30°.

#### 3.3.7. Contact Angle Analysis

For this analysis, films of the TPSs and TPSg blends were obtained in a press at a temperature of 110 °C and a pressure of 2 tons. The contact angle was measured using a Ramé-Hart Model 250 goniometer with an optical system with which the interaction of water (2 μL) with the surface of the films of the samples was observed. The image was captured after 60 s and the analysis was carried out with Image J software. Three measurements were made for each sample to take an average of the measurements.

## 4. Conclusions

This study has shown that the hydrophilic properties of the thermoplastic starch modified in the molten state with organic acids prove favorable.

The gelatinization process was reached in better effectiveness in TPS functionalized with lactic acid, whether the plasticizer was sorbitol or glycerol. This was observed in lower values of contact angle and enthalpy energy of the acid.

The addition of lactic acid promoted lower values of the loss modulus, storage modulus, and complex viscosity, which means lower waste energy in the transition to viscous from elastic state making this functionalization (sorbitol-glycerol with lactic acid) more feasible for processing scaling.

Taking into account the results presented for each of the organic acids evaluated in this research, the need to look for a synergy is observed to obtain the advantages of using each of them by modifying achira starch with a mixture of both acids during the same process. These results are very promising for the continuity of research in the search for new biodegradable polymers to obtain natural-origin packaging materials.

## Figures and Tables

**Figure 1 molecules-24-04433-f001:**
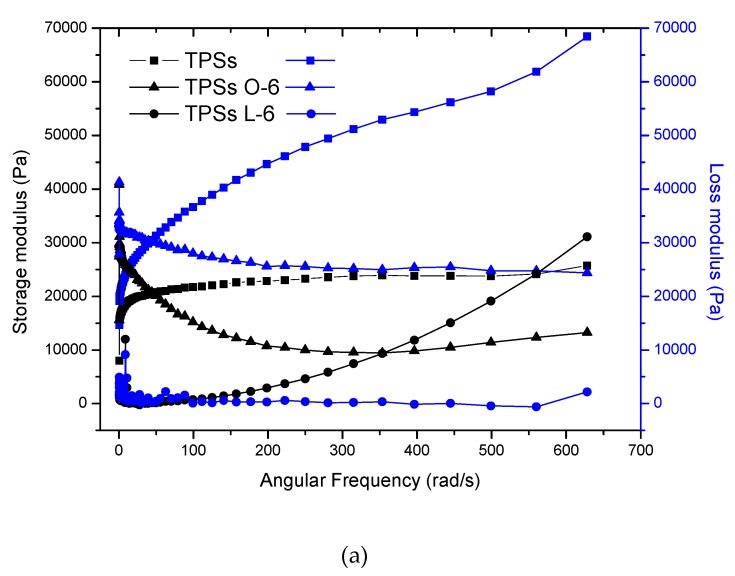
Rheograms of TPS samples with sorbitol and different acids. (**a**) Storage modulus (G’) and loss modulus (G’’); and (**b**) complex viscosity.

**Figure 2 molecules-24-04433-f002:**
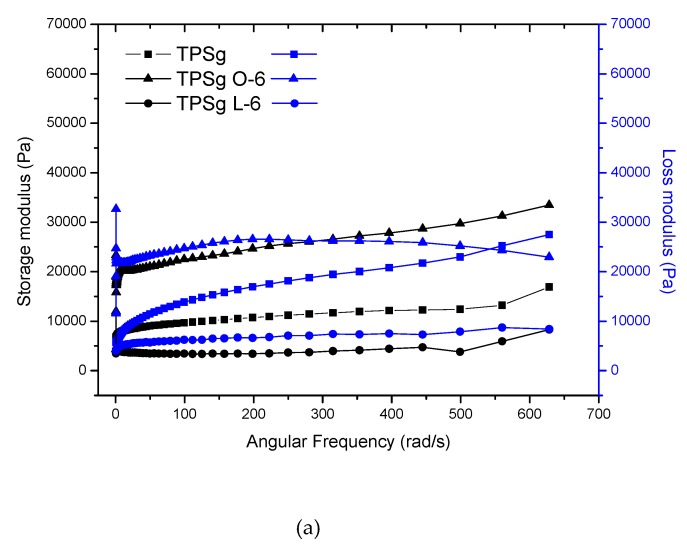
Rheograms of TPS samples with glycerol and different acids. (**a**) Storage modulus (G’) and loss modulus (G’’); and (**b**) complex viscosity.

**Figure 3 molecules-24-04433-f003:**
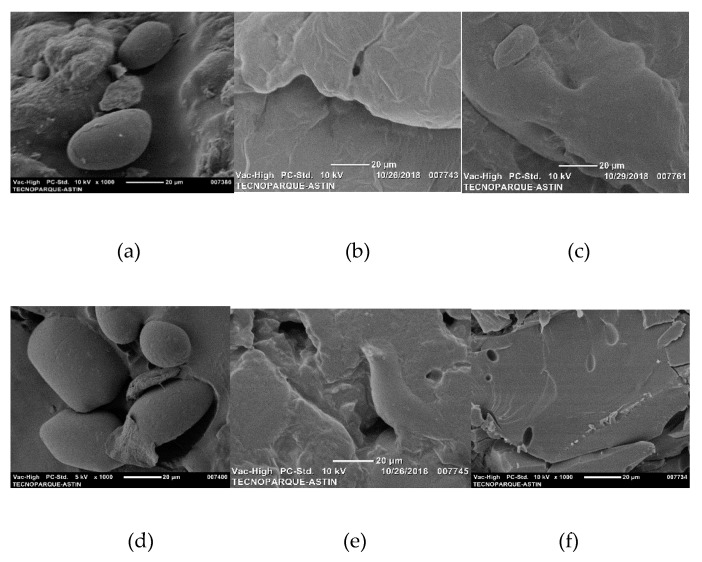
Micrographs obtained by SEM of TPS with 6% acid agent. (**a**). TPSs, (**b**). TPSs-O6, (**c**). TPSs L-6, (**d**). TPSg, (**e**) TPSg O-6, (**f**) TPSg L-6.

**Figure 4 molecules-24-04433-f004:**
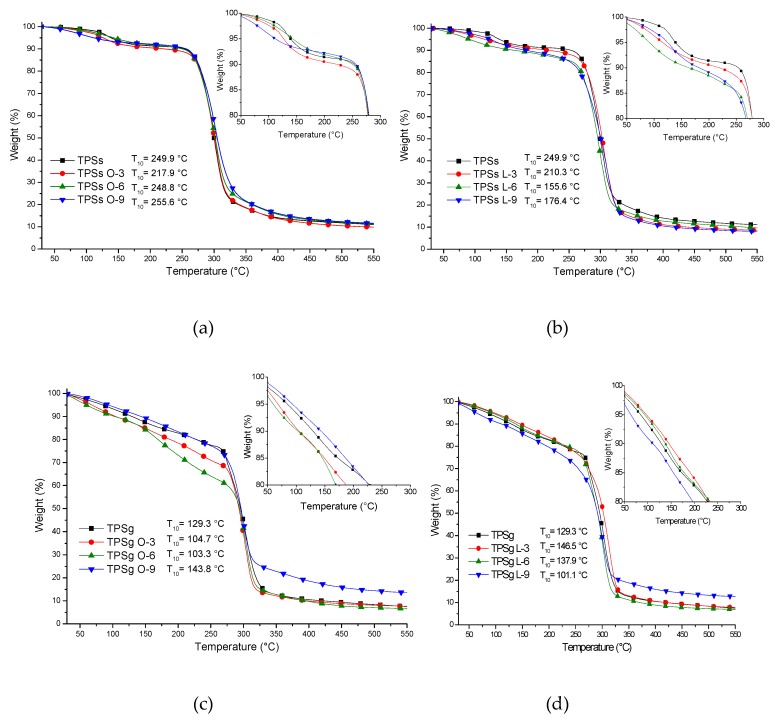
The TGA thermogram for TPS samples. (**a**) TPSs-O, (**b**) TPSs-L, (**c**) TPSg-O, (**d**) TPS-L.

**Figure 5 molecules-24-04433-f005:**
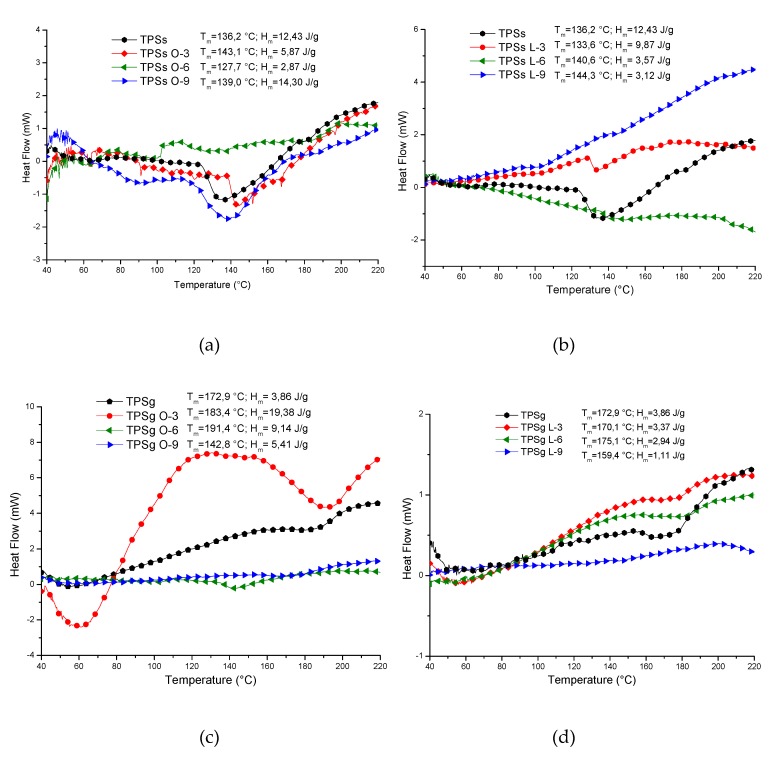
DSC curves for TPS samples; (**a**) TPSs-O, (**b**) TPSs-L, (**c**) TPSg-O, (**d**) TPSg-L.

**Figure 6 molecules-24-04433-f006:**
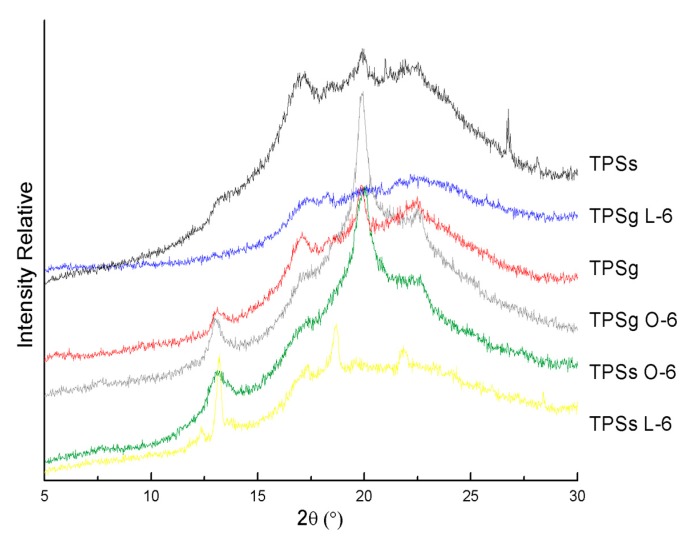
Diffractograms of plasticized achira TPS with organic acids.

**Figure 7 molecules-24-04433-f007:**
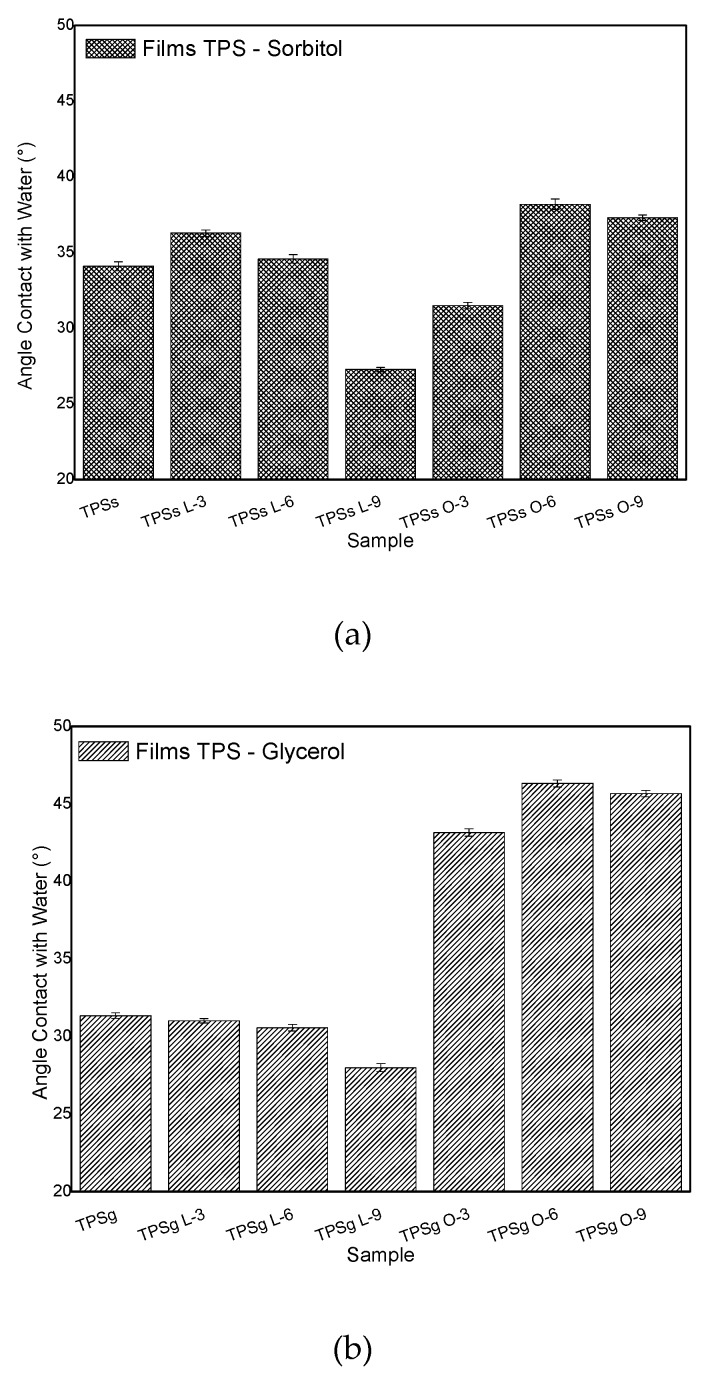
Contact angle of achira TPS. (**a**) Samples obtained with sorbitol. (**b**) Samples obtained with Glycerol.

**Table 1 molecules-24-04433-t001:** Torque rheometry results of the TPS mixtures with oleic acid (O) and lactic acid (L) plasticized with sorbitol (s) and glycerol (g).

Sample	Torque _max_ (Nm)	Torque _min_ (Nm)	T _min_ (°C)	Energy (kJ)
TPSs	12.8	10.3	109.2	57.4
TPSs O-3	9.9	9.4	99.6	53.7
TPSs O-6	8.3	8.0	95.1	43.0
TPSs O-9	20.0	13.7	105.8	80.4
TPSs L-3	6.2	3.2	111.7	4.7
TPSs L-6	9.1	5.9	89.0	35.1
TPSs L-9	11.5	7.3	100.2	24.3
TPSg	47.0	31.2	93.6	166.3
TPSg O-3	9.1	8.8	111.0	48.2
TPSg O-6	11.4	9.5	113.5	44.3
TPSg O-9	32.5	30.1	98.5	156.2
TPSg L-3	7.1	0.4	98.6	4.7
TPSg L-6	10.8	2.3	110.0	18.3
TPSg L-9	38.5	13.5	84.6	119.4

**Table 2 molecules-24-04433-t002:** Proportions of starch, plasticizing, and acid agent used for the preparation of the thermoplastic starch.

Samples	Starch (wt.%)	Plasticizing (wt.%)	Acid Agent (wt.%)
TPSs	70.0	30.0	0
TPSs O-3	67.9	29.1	3
TPSs O-6	65.8	28.2	6
TPSs O-9	63.7	27.3	9
TPSs L-3	67.9	29.1	3
TPSs L-6	65.8	28.2	6
TPSs L-9	63.7	27.3	9
TPSg	70.0	30.0	0
TPSg O-3	67.9	29.1	3
TPSg O-6	65.8	28.2	6
TPSg O-9	63.7	27.3	9
TPSg L-3	67.9	29.1	3
TPSg L-6	65.8	28.2	6
TPSg L-9	63.7	27.3	9

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
