# Peer review of "Rheological, Thermal, Superficial, and Morphological Properties of Thermoplastic Achira Starch Modified with Lactic Acid and Oleic Acid"

_molecules, 2019, doi:10.3390/molecules24244433_

Round 1

Reviewer 1 Report

Authors omitted reviewer's suggestion about necessity of mechanical properties evaluation (tensile strenth, Young modulus and extension at break) for modified TPS films. Moreover, there are still some shortcomings in citation of references such as: (i) names of journals written sometimes with low characters, (ii) numbers of issues given in some cases are not necessary, (iii) lack of full bibliographic data: ref. 12 and 41.

So, taking above into account authors should answer the above comments before decision about manuscript acceptation.

Author Response

Elsa Wang

Assistant Editor, MDPI

Molecules

Please find attached the revised version of the manuscript entitled “Rheological, thermal, superficial and morphological properties of thermoplastic Achira starch modified with lactic acid and oleic acid” by Carolina Caicedo*, Rocio Yaneli Aguirre Loredo, Abril Fonseca García, Omar Hernan Ossa, Aldo Vázquez Arce, Heidy L. Calambás Pulgarin and Yenny Patricia Avila Torres.

This revised version has been prepared taken into consideration the comments kindly offered by the reviewers. In what follows we comment on the reviewers remarks and describe the changes made to the original version.

Reviewer 1

1) Recommendation: Authors omitted reviewer's suggestion about necessity of mechanical properties evaluation (tensile strength, Young modulus and extension at break) for modified TPS films.

Comments: In this study it was not considered to evaluate the mechanical properties of the prototype in solid state (film), however, the mechanical properties of the fluid were evaluated in order to know details about the processability. We will consider this recommendation for a later study in which a prototype with the optimal mixture derived from this evaluation is obtained considering the synergy between starch:plasticizer:acidic agent.

2) Recommendation: Moreover, there are still some shortcomings in citation of references such as: (i) names of journals written sometimes with low characters, (ii) numbers of issues given in some cases are not necessary, (iii) lack of full bibliographic data: ref. 12 and 41.

Comments: In relation to these comments (i) Journals name were modified due to every word begins in the capital letter, (ii) EndNote software was used to insert the citations in the article. However, every citation was download, but every article had a specific format. The correction was done. (iii) ref 12 and 41 were completed.

We appreciate all the suggestions made by the reviewers. In this new version we attended the best we could all comments.

Sincerely,

Authors

Manuscript ID: molecules- 655090

Reviewer 2 Report

The authors have provided a new version, in which they have improved the quality of this manuscript. Several items have now been clarified, but there are still issues requiring further editing/clarification:

- Major English language edition is still needed. Especially the new text in red color needs checking.

- Abstract is rewritten, but I think it is now too long. The abstract should be a total of about 200 words maximum, now it is over 300.

- The discussion of plasticizer amount in lines 117-125 should be in the results section.

-Table 2: Please use the same amount of decimals in all values

-Section 3.2 Rheological analysis: Please discuss why TPSs L-6 has abnormal complex viscosity. Please correct the caption of Figure 1.

- Quality of TGA and DSC curves should be improved. Now the lines are very dull and thin.

-The authors have shown the DSC curves with melting temperatures in Figure 5, but they have not discusses these melting temperatures in the text. Are the reported temperatures real melting temperatures or softening temperatures? At least in case of TPSg L-samples, the DSC curves are quite straight.

Author Response

Elsa Wang

Assistant Editor, MDPI

Molecules

Please find attached the revised version of the manuscript entitled “Rheological, thermal, superficial and morphological properties of thermoplastic Achira starch modified with lactic acid and oleic acid” by Carolina Caicedo*, Rocio Yaneli Aguirre Loredo, Abril Fonseca García, Omar Hernan Ossa, Aldo Vázquez Arce, Heidy L. Calambás Pulgarin and Yenny Patricia Avila Torres.

Reviewer 2

The authors have provided a new version, in which they have improved the quality of this manuscript. Several items have now been clarified, but there are still issues requiring further editing/clarification:

Recommendation: Major English language edition is still needed. Especially the new text in red color needs checking.

Comments: Style correction was performed in the language.

Recommendation: Abstract is rewritten, but I think it is now too long. The abstract should be a total of about 200 words maximum, now it is over 300.

Comments: The abstract was reduced to 199 words.

Recommendation: The discussion of plasticizer amount in lines 117-125 should be in the results section.

Comments: The indicated correction was made.

Recommendation: Table 2: Please use the same amount of decimals in all values

Comments: The values were identified and these were corrected.

Recommendation: Section 3.2 Rheological analysis: Please discuss why TPSs L-6 has abnormal complex viscosity.

Comments: We have added the following text in page 6 to complement the rheological explanation. “The reduction in the viscosity of the TPS corresponds to the dissociation of the interactions and disentanglement of the double helices of amylose and amylopectin. On the other hand, at angular frequency of ~100 rad/s, a considerable increase in the η* value was observed for all TPS with lactic acid (shear thickening). This behavior has been evidenced in tapioca starch samples possibly given by the formation of a structure of the elastic network, which occurs when the material is exposed to external stresses due to a large increase in the dispersion's storage modulus[43]. The formation of this structure can be produced by chemical or physical crosslinking, to the complexation reaction between amylose and lipids, or the physical entanglement of the high molecular weight polysaccharides[44]. The angular frequency value, in which the steep increase appears in η* did not show a trend with the proportions of starch and acid”.

Please correct the caption of Figure 1.

Comments: The indicated correction was made

Recommendation: Quality of TGA and DSC curves should be improved. Now the lines are very dull and thin.

Comments: Lines and symbols were used to improve the quality of the thermogram figures.

Recommendation: The authors have shown the DSC curves with melting temperatures in Figure 5, but they have not discusses these melting temperatures in the text. Are the reported temperatures real melting temperatures or softening temperatures? At least in case of TPSg L-samples, the DSC curves are quite straight.

Comments: We have added the following text in page 11 to complement the explanation of the results of the DSC. “It is important to mention that in case of a correct reaction between starch and plasticizer, the peak indicating the melting point of the pure plasticizer (17 °C and 96 °C for glycerol and sorbitol, respectively) should not appear in the DSC analysis since this constituent would become part of the structure of the TPS. As we can see in some cases the TPSg samples record a large endothermic peak between 40 °C and 100 °C representing the evaporation of glycerol that was not incorporated. Thus, in the joint analysis of TGA and DSC, the loss of mass is evidenced by partial volatilization of the plasticizer that fails to interact. In the TPSs, fusion transitions are observed between 120 °C and 190 °C, which varied depending on the proportions of lactic acid but independent for oleic acid, probably due to lower efficiency in incompatibility, nature of fatty acids versus polar molecules.”

We appreciate all the suggestions made by the reviewers. In this new version we attended the best we could all comments.

Sincerely,

Authors

Manuscript ID: molecules- 655090

Round 2

Reviewer 2 Report

I am satisfied with the changes introduced and have no further comments.